# Quantifying global colonization pressures of alien vertebrates from wildlife trade

Yiming Li [1,2,3] ✉, Tim M. Blackburn [4,5], Zexu Luo[2,3], Tianjian Song[2,3], Freyja Watters[6], Wenhao Li[2,3], Teng Deng[2,3], Zhenhua Luo[7], Yuanyi Li[1], Jiacong Du[1], Meiling Niu[1], Jun Zhang[1], Jinyu Zhang[1], Jiaxue Yang[1] & Siqi Wang[2,3]

The global trade in live wildlife elevates the risk of biological invasions by increasing colonization pressure (the number of alien species introduced to an area). Yet, our understanding of species traded as aliens remains limited. We created a comprehensive global database on live terrestrial vertebrate trade and use it to investigate the number of traded alien species, and correlates of establishment richness for aliens. We identify 7,780 species involved in this trade globally. Approximately 85.7% of these species are traded as aliens, and 12.2% of aliens establish populations. Countries with greater trading power, higher incomes, and larger human populations import more alien species. These countries, along with island nations, emerge as hotspots for establishment richness of aliens. Colonization pressure and insularity consistently promote establishment richness across countries, while socio-economic factors impact specific taxa. Governments must prioritize policies to mitigate the release or escape of traded animals and protect global biosecurity.

The wildlife trade, encompassing both legal and illegal activities[1–3], represents a meaningful human commodity and a significant contribution to the global economy[4,5], but ranks among the foremost threats to global biodiversity and environmental security[6]. Wildlife trade represents the sale of non-domesticated animals, plants or fungi, whether taken from their natural environment or raised in captivity. This can include both live or dead individuals and their body parts. The selling of protected wildlife or their parts in contravention of local, national, or international laws is known as illegal wildlife trade. Traded live wildlife includes native species – sold within the range where they naturally occur – and alien species – traded beyond the borders of their native range. The latter present major challenges to global biosecurity[7], as they can escape or be released into the wild and establish viable populations, posing threats to species persistence[8], and emerging disease risks[9]. The trade in live wildlife, both legal and illegal, has grown dramatically over recent decades as increasing

human populations and incomes have fostered demand for exotic pets[10,11], which can be supplied by improved international transport capacity and rapid growing online trade[12–14]. The volume and number of alien species in trade have increased concomitantly: millions of wild-caught or captive-bred live animals are traded annually as pets, or for zoos, food, and other uses[10], including many of the most notorious invasive alien species (e.g. Red-eared Slider *Trachemys scripta elegans*, African Clawed Frog *Xenopus laevis*, Burmese Python *Python bivittatus*)[15,16].

Previous studies have addressed the impacts of the wildlife trade on species persistence and abundance in their native distributions[10,11,13,14,17–20], but invasion risks from alien species in trade have received comparatively less attention[12,15,21]. Studies to date have focused on a single taxon (e.g., birds or reptiles), or a specific human use (e.g., pets), and on regional scales[15,21,22]. The key questions of how many species are involved in the live wildlife trade as aliens

[1]School of Life Sciences, Institute of Life Sciences and Green Development, Hebei University, Baoding 071002, China. [2]Key Laboratory of Animal Ecology and Conservation Biology, Institute of Zoology, Chinese Academy of Sciences, 1 Beichen West Road, Chaoyang 100101 Beijing, China. [3]University of Chinese Academy of Sciences, 100049 Beijing, China. [4]Centre for Biodiversity and Environment Research, University College London, Gower Street, London WC1E 6BT, UK. [5]Institute of Zoology, Zoological Society of London, Regent's Park, London NW1 4RY, UK. [6]Invasion Science & Wildlife Ecology Lab, University of Adelaide, Adelaide, SA, Australia. [7]School of Life Sciences, Central China Normal University, NO.152 Luoyu Road, Wuhan 430079 Hubei, China. ✉e-mail: liym@ioz.ac.cn

outside their native ranges, and to what extent these aliens establish viable populations worldwide, remain to be resolved. These knowledge gaps are becoming increasingly urgent to fill in response to calls for action related to strengthened wildlife trade surveillance, with wildlife trade at global scale growing and likely unsustainable, especially post the COVID-19 pandemic[23,24].

The number of alien species that establish viable populations in an area, here termed *establishment richness*, is determined by three variables: the number of alien species introduced (colonization pressure), the number of individuals of each species introduced (propagule pressure), and the probability that a founding individual leaves a surviving descendant (lineage survival probability)[25]. Colonization pressure is the number of species with the opportunity to establish a viable alien population (i.e. those that are introduced), while propagule pressure and lineage survival probability determine which, and how many, of the introduced alien species actually do establish. Socio-economic factors and environmental conditions are likely to affect these variables, and hence numbers of established alien species[15,16,26,27]. Lineage survival probability will depend on abiotic and biotic conditions, such as climate match and native species richness[28]. While islands tend to have higher establishment richness of alien species than mainland regions[29,30], it is currently disputed whether this is due to higher colonization or propagule pressures, or natural features of islands, such as more amenable climates or lower biotic resistance from the relatively impoverished biotas found on islands. Colonization pressure data are key to distinguishing these effects, yet there has to date been no attempt to disentangle the effects of colonization pressure and other factors on global spatial patterns of establishment richness along a specific invasion pathway. Such an attempt would provide key information for preventing the introduction of alien species and identifying regions with high invasion risks associated with wildlife trade.

In this study, we have compiled a comprehensive global live (terrestrial) vertebrate trade database (GLVTD; see Methods, Supplementary Data 1). The GLVTD catalogs species from four vertebrate groups – mammals, birds, reptiles, and amphibians, that are involved in various aspects of the live wildlife trade. The database includes species that are sold through online trade and physical stores (OTAPS) for pets or other uses, species that are kept in zoos, and the countries or regions that imported or exported this wildlife. We define alien species in the GLVTD as those that are traded beyond the borders of their native range[31], regardless of whether they have established alien populations or not. We use this database (i) to demonstrate the geographical distribution of colonization pressure for alien vertebrates in trade across countries and taxa, and their associations with socio-economic factors; (ii) to identify hotspots of establishment richness for alien species and the contributions of colonization pressure, socio-economic factors and climate conditions to this richness; and (iii) to quantify flows of all and established alien species between native regions and recipient regions.

## Results

### Global colonization pressures of alien vertebrates in trade

We collated data on the species involved in the global trade of live, terrestrial vertebrates from multiple sources, including the Convention on International Trade in Endangered Species of Wild Fauna and Flora (CITES) Trade Database (2371 species), the United States Fish and Wildlife Service's (USFWS) Law Enforcement Management Information System (LEMIS, 3908 species), the International Species Information System (ISIS, 3116 species), and various online trade platforms and physical stores (OTAPS, 5053 species) in order to compile a comprehensive, global database of live (terrestrial) vertebrate wildlife trade, referred to as GLVTD. The GLVTD includes 7780 unique species involved in the live vertebrate trade worldwide (Fig. 1, Fig S1 a–d in Supplementary Information, and Supplementary Data 1). Approximately 45.1% ($n = 3508$) of the species were unique to individual datasets, while 54.9% overlapped between two, three or four datasets. According to the International Union for Conservation of Nature

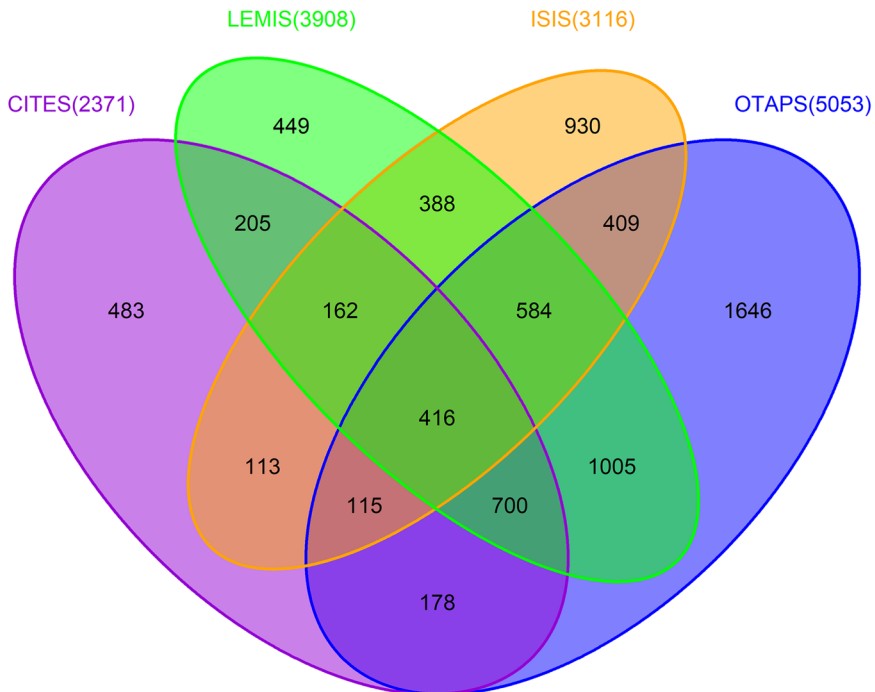

**Fig. 1 | Venn Diagram of species assembled (total 7780 species) from different data sources for the live wildlife trade based on GLVTD.** The number of species contained within each data source is given in parentheses. CITES: CITES Trade Database; LEMIS: the United States Fish and Wildlife Service's (USFWS) Law Enforcement Management Information System; ISIS: International Species Information System; OTAPS: the dataset obtained from online trade and physical stores. The numbers in the diagram indicate the number of species in different sets in a data source or the intersections among multiple data sources. The figure is created by the VennDiagram package in R (Venn Diagram in Supplementary Code 1).

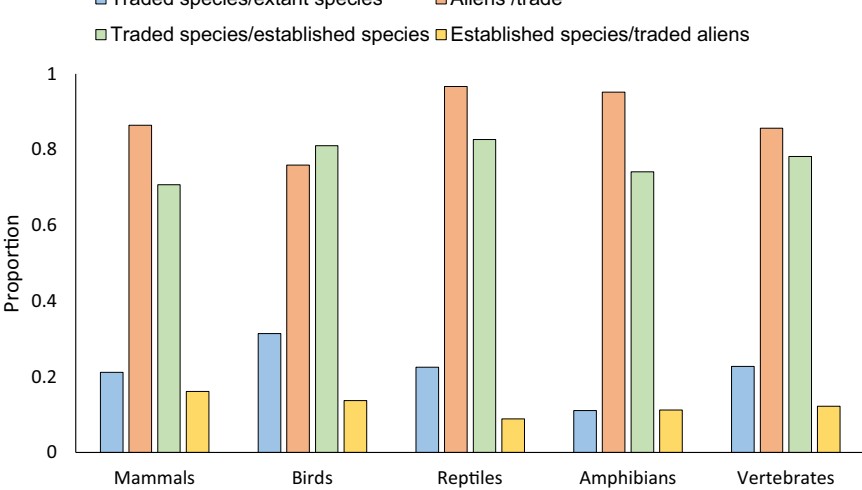

**Fig. 2 | Proportions of traded alien species and establishment in extant vertebrates.** The blue bar represents the proportion of traded species among all extant species; the red bar indicates the proportion of alien species among all traded species; the green bar shows the proportion of traded species among all established species; the yellow bar displays the proportion of established species among all traded alien species. Source data are provided as a Source Data file. The figure is created by Excel.

(IUCN) taxonomy, these species account for 22.7% of the 34,285 extant terrestrial vertebrate species listed on the IUCN Red List (Fig. 2, shown in blue). They include 1247 species from 125 families of mammals, 3451 species from 207 families of birds, 2278 species from 82 families of reptiles, and 804 species from 53 families of amphibians (Supplementary Data 1).

We matched the list of countries or regions (based on global administrative areas at the country or region level) where a species is traded with its native geographic range from the IUCN Red List data for each species (Methods), and identified alien species as those that are traded in a country or region where they do not naturally occur. While there were 14 species lacking data on geographical range on the Red List (Table S1 in Supplementary Information), all traded species had range data. We identified 6664 vertebrate species that were traded outside their native range as aliens, including 1078 species of mammals, 2619 birds, 2202 reptiles and 765 amphibians. Approximately 65.9% of these species (4392/6664) were also traded within their native ranges. Overall, 85.7% (6664/7780) of the terrestrial vertebrate species in global trade were traded as aliens somewhere (Fig. 2, in red): this includes 86.4% of mammals, 75.9% of birds, 96.7% of reptiles, and 95.1% of amphibians involved in global trade. Conversely, only 14.3% (1116 /7780) of species were traded solely within their native range.

Each of 193 countries had records of alien species in trade (Fig. 3a, Fig S2a–d, Supplementary Data 2). The number of traded alien species ranged from one species in Micronesia to 4600 species in the United States, with an average of 425.7 ± 637.1 species/country. Particularly high numbers of alien vertebrate species have been imported to countries in North America (Canada (2713 species) and the United States), and Western Europe, such as Germany (3171 species) and Great Britain (2731 species). Similar patterns were observed across vertebrate classes (Fig S2 a–d), with strong correlations in alien trade richness across countries (Table S2, r ≥ 0.782, p < 0.01 for all pairs).

Sampling bias in alien species records existed across countries or regions, with lower sampling effort especially on regional scales due to the limited coverage of the CITES Trade Database, and an absence of comprehensive online trade surveys (see Methods). Nations with upper middle or high incomes tended to have more open economies and invest greater resources in biodiversity conservation[32,33], resulting in more comprehensive data on wildlife trade compared to nations with lower middle or low incomes. Furthermore, all countries with upper middle or high income were CITES parties and had relatively complete records of CITES-restricted species. We therefore examined patterns in the number of alien vertebrate species associated with socio-economic factors across countries with upper middle or high income (Methods). We found that the number of alien species traded in a country increased (estimate >0) with the amount of commercial trade (total value of import and export goods), human population size and per capita GDP (GDPpc) (p < 0.001 for each factor, Table S3). The amount of commercial trade accounted for 79.8% of the variation in the number of traded alien species in univariate analysis, compared to 57% by human population size and 7.1% by GDPpc.

On average, alien species accounted for 83.9% of species richness in trade for vertebrates within a country (Fig. 4), ranging from 79.1% in birds to 89.8% in reptiles. The average number of alien species in trade was 5.21 times higher than the number of native species across countries, indicating a substantial dominance of alien species in the live terrestrial vertebrate wildlife trade. This dominance was repeated in all vertebrate groups (Fig. 4, Paired t test, p < 0.001 for all, Table S4).

## Contributions of colonization pressure and other factors to establishment richness

We identified 1041 vertebrate species with established alien populations, of which 814 were involved in the live wildlife trade. This included 174 species of mammals, 359 birds, 195 reptiles and 86 amphibians (Supplementary Data 3). Traded species with established populations accounted for 12.2% of alien vertebrates in trade (as shown in yellow in Fig. 2; 16.1% of mammals, 13.7% of birds, 8.9% of reptiles, and 11.2% of amphibians). Traded species comprised 78.2% of all established vertebrate species, ranging from 70.3% for mammals to 82.6% for birds (as shown in green in Fig. 2).

Hotspot countries for the establishment richness of traded alien species included the United States (288 species), Australia (118 species), Spain (89 species) and France (77 species), as well as several island nations such as New Zealand (87 species), Japan (85 species) and Great Britain (75 species) (Fig. 3b and Fig S3 a–d, Supplementary Data 4). Emerging countries in the global economy, like Brazil, South Africa, Mexico, Russia and China, had moderate establishment richness. Establishment richness of alien species in trade was again correlated between taxonomic groups across countries (Table S2, r ≥ 0.207, p < 0.01 for all pairs).

Established species were traded in more countries compared to unestablished species for all taxa (p ≤ 0.003) and in more areas than

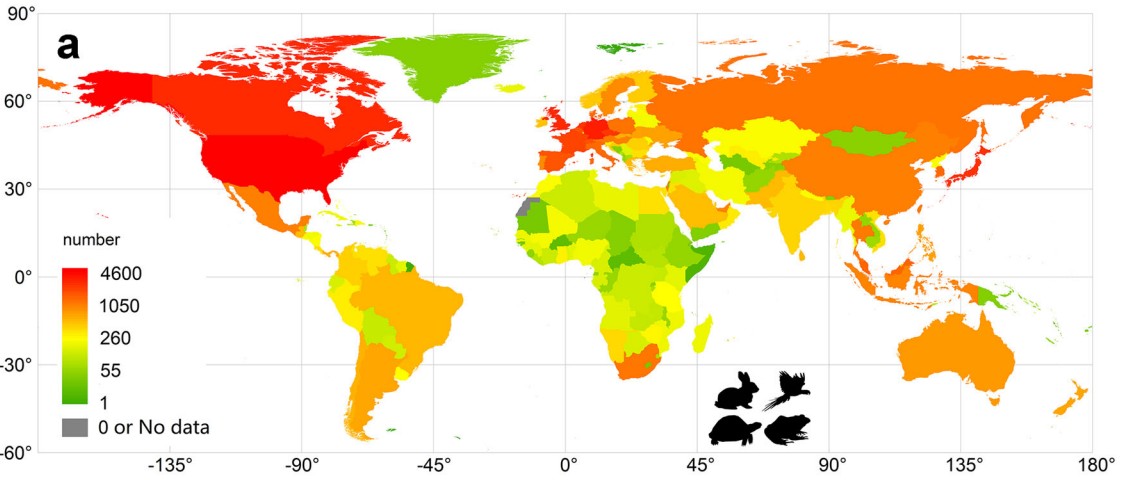

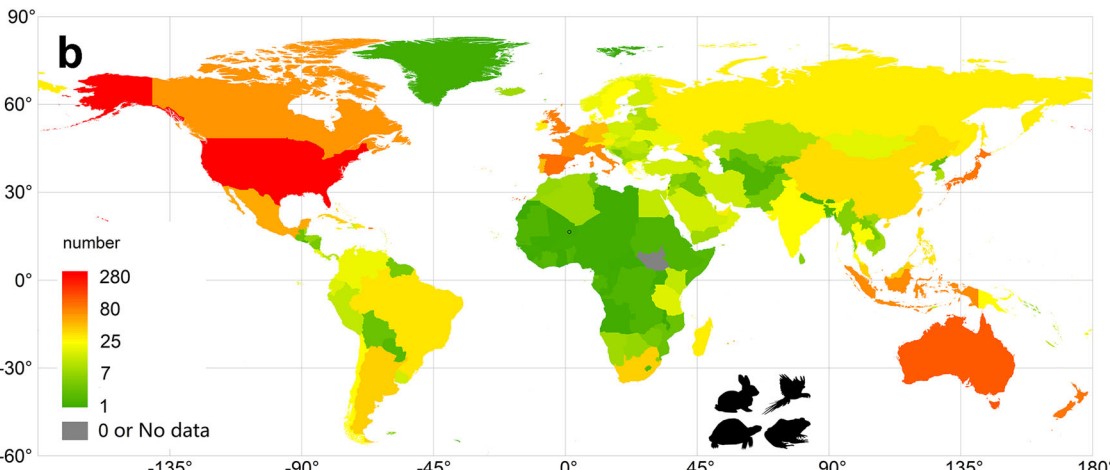

**Fig. 3 | The geographical distribution of alien vertebrate richness in trade (included established and unestablished species, total 6,664) and establishment richness (total 814 species) across the globe (data from GLVTD).** The figure is created by ArcGIS; (**a**) alien vertebrate richness (also see Supplementary Data 2 for original data); (**b**) establishment richness (Supplementary Data 4). Versions with alternative colour schemes are provided with Fig S2e–i and Fig S3e–i in Supplementary Information.

unestablished species for all taxa ($p \leq 0.011$) except amphibians (Table S5, Fig S4, 5). For example, established species were traded, on average, in 1.18 times more countries compared to unestablished species for mammals (established species = $18.3 \pm 21.2$ countries vs unestablished species = $15.5 \pm 23.6$ countries). Similarly, for birds, established species were traded in 2.62 times more countries ($28.7 \pm 37.0$ vs $11.0 \pm 19.4$). For reptiles, the ratio was 2.33 times ($23.4 \pm 25.6$ vs $10.1 \pm 14.1$), while for amphibians it was 1.34 times ($9.5 \pm 13.0$ vs $7.1 \pm 9.7$) (Fig S4).

We used multimodel inference and information theory (Akaike's Information Criterion corrected for small sample sizes, AICc)[34] (see Methods) to quantify the relative contributions of colonization pressure (the number of alien species in trade as a measure of colonization pressure), socio-economic factors, and environmental conditions to the establishment richness of traded alien species across upper middle and high income countries (100 nations). This approach makes a more reliable inference of the relative importance of predictors, compared to any single model, by including a group of models and merging model uncertainty[34,35]. Conditional averaging based on linear mixed models showed that colonization pressure and insularity were consistent predictors of establishment richness for each group (Table 1). Establishment richness in a country increased (estimate >0) with

colonization pressure and insularity for all taxa, with area, GDPpc, and population density for birds and amphibians, and with sampling effort for mammals. Furthermore, establishment richness increased with temperature for reptiles but decreased (estimate <0) with temperature for mammals (Table 1).

Colonization pressure and insularity were also included in all the highly supported models (i.e., $\Delta$AICc $\leq 2$) for each group (Table S6–9). Fixed factors explained 62.4–67.5% of the variation in establishment richness ($R^2m$) for mammals (Table S6), 59.8–60.9% for birds (Table S7), 29.7–31.5% for reptiles (Table S8), and 28.96–39% for amphibians (Table S9).

## The networks of flows of alien species and established alien species in trade

Every economic region worldwide imported and exported alien vertebrates from or to other regions, with interregional exchange dominating the flows of species (Fig. 5a). North America, Europe, and South and East Asia imported the largest numbers of species, while South and East Asia, Africa, South America and North America were the main export regions. For established species, North America, Europe, South and East Asia and Oceania were the main recipients (Fig. 5b), while South and East Asia, Africa, Europe, and North America were the main

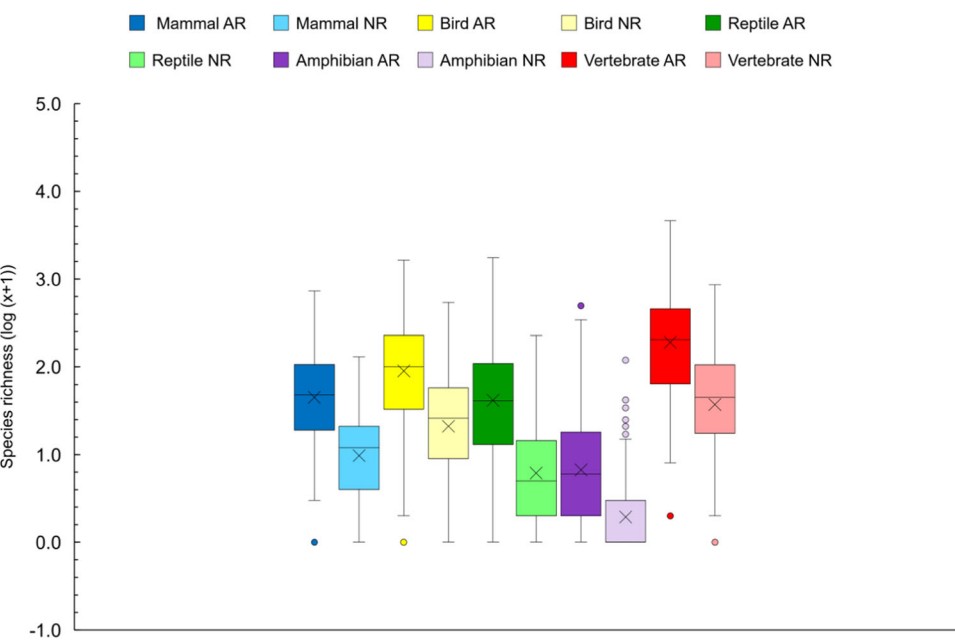

**Fig. 4 | Box plot of alien richness versus native richness in live wildlife trade across 193 countries (*n* = 193).** AR and NR represent alien species richness and native species richness, respectively. The black line and X inside the box indicate the median and mean, respectively. The bottom and top borders of the box represent the first and third quartiles. The vertical dotted lines outside the box represent the upper and lower limits. The outliers are represented as dots. Source data are provided as a Source Data file. The figure is created by Excel.

**Table 1 | Predictors of establishment richness of traded alien vertebrates across countries with upper middle income or high income**

| Taxa | Mammals | | Birds | | Reptiles | | Amphibians | |
| --- | --- | --- | --- | --- | --- | --- | --- | --- |
| Predictors | Estimate | Pr(>|z|) | Estimate | Pr(>|z|) | Estimate | Pr(>|z|) | Estimate | Pr(>|z|) |
| Intercept | 0.174 | 0.707 | −2.763 | **0.000** | −0.276 | 0.647 | −0.822 | 0.460 |
| Area | 0.006 | 0.911 | **0.248** | **0.000** | 0.112 | 0.229 | **0.171** | **0.007** |
| Population density | 0.063 | 0.227 | **0.305** | **0.000** | 0.129 | 0.190 | **0.215** | **0.015** |
| GDPpc | 0.104 | 0.169 | **0.432** | **0.000** | 0.161 | 0.195 | **0.221** | **0.022** |
| Colonization pressure | **0.358** | **0.000** | **0.210** | **0.012** | **0.330** | **0.000** | **0.219** | **0.001** |
| Insularity | **0.234** | **0.032** | **0.285** | **0.000** | **0.236** | **0.002** | **0.194** | **0.012** |
| Mean temperature | **−0.028** | **0.000** | 0.005 | 0.153 | **0.014** | **0.009** | 0.000 | 0.999 |
| Mean precipitation | 0.090 | 0.275 | −0.088 | 0.174 | 0.167 | 0.125 | 0.141 | 0.119 |
| Congeneric richness | 0.276 | 0.071 | −0.098 | 0.379 | −0.051 | 0.243 | 0.104 | 0.184 |
| Sampling effort | **0.545** | **0.004** | 0.124 | 0.434 | 0.364 | 0.078 | 0.226 | 0.213 |

The table summarizes the standard estimates and probabilities of regression coefficients based on conditional averaging ($2^9 = 512$ models) for linear mixed models with the relationship between the number of established traded alien species in a country as the response variable and combinations of 9 factors as predictors (fixed effects) across 100 countries (*n* = 100). Biogeographical realm was included as a random effect. Significant results are marked in bold type.

donors. Intraregional exchange was relatively more frequent for established alien species than for species in trade in general (Fig. 5a, b). Patterns were largely similar across taxa (Figs. S6a–d, S7a–d).

## Discussion

Our analyses quantify the numbers of alien species in live wildlife trade at the global scale and country level, and identify geographic patterns of traded alien species and drivers of establishment richness for alien species globally. We find that, globally, most species (75.9–96.7%, depending on class) in live wildlife trade are traded outside their native range, and hence are aliens, and that aliens comprise much higher proportions (79.1–89.8%) of species in trade in each country than do natives. These findings suggest that aliens dominate species richness in the live wildlife trade. Countries with larger human populations, higher incomes, or larger trading powers have larger numbers of alien vertebrate species in trade, at least for upper middle income or high

income countries. Colonization pressure and insularity are consistently strong predictors of establishment richness for every vertebrate taxon.

Large human populations are associated with a strong demand for, and commercial consumption of, live wildlife. Countries with greater trading power may have more opportunities or pathways to access different source pools of species[36], and therefore have a higher number of imported alien species. GDPpc partly reflects the import volume of alien wildlife and release frequency. With increasing living standards (and GDPpc), the market for exotic pets (e.g., species without a long history of domestication) expands and pet ownership grows[15,37,38]. This likely increases pet import volumes and promotes occasional or intended releases, and hence increases propagule pressure, the key driver of alien population establishment[7,39]. Previous studies have generally used socio-economic factors such as human population size, GDPpc and trade volume as surrogates for

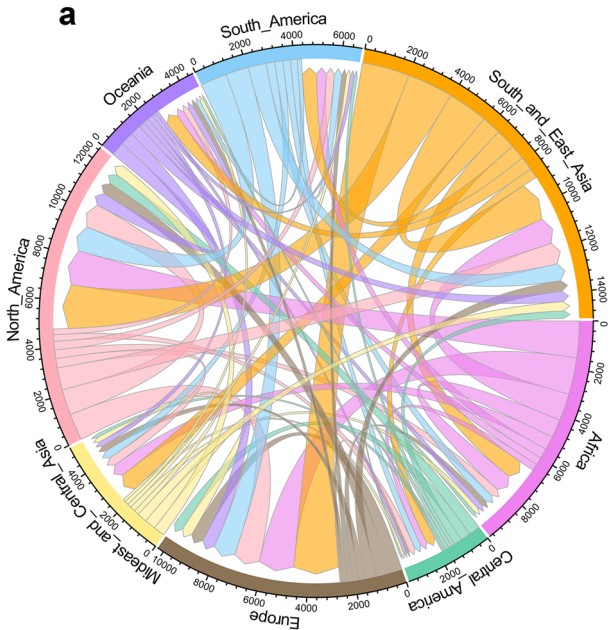

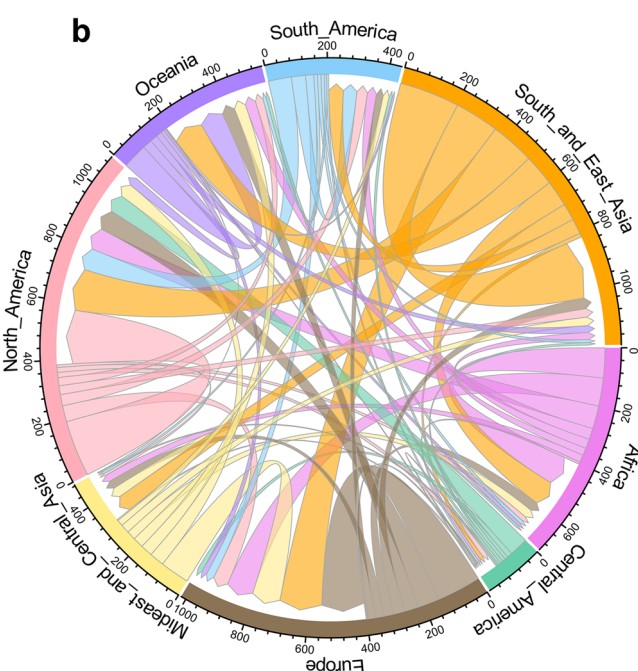

**Fig. 5 | Network analysis of the global flows of traded alien vertebrates (total 6,664 species) and established alien vertebrates (total 814 species) among 8 economic regions.** A unique colour indicates a region where species are native. The ribbons show the flows of species linked from native (no gaps) to alien regions (with gaps), with the size of ribbons representing the volume of species flow (the same species may be counted multiples due to its origination from multiple regions or its trade in multiple regions). The tick marks on unique colour segments indicate absolute number of species that are imported or exported from a region. The figure is created by the dplyr, circlize and reshape2 packages in R (Network analysis in Supplementary Code 1). **a** Traded alien vertebrates; (**b**) established alien vertebrates.

colonization pressure[29,33,40,41]. The results of this study confirm positive correlations between colonization pressure and these socio-economic factors. The evidence provided by the number of alien species in trade for each taxon supports the hypothesis that colonization pressure is a fundamental determinant of spatial variation in the establishment

richness of aliens in an area[25,28]. This relationship, which has rarely been examined at a large scale, suggests that the level of colonization pressure, as indicated by the number of alien species involved in trade, plays a significant role in shaping the establishment richness of alien species in different regions.

The relative higher establishment richness observed in island countries compared to mainland countries across taxonomic groups indicates that island countries are more susceptible to invasion pressure from alien vertebrates. This result confirms an island effect while accounting for the confounding effect of colonization pressure. It most likely arises from low biological resistance (e.g., low predator, competitor, or parasite pressures) because of reduced biodiversity or increased ecological naiveté of native insular communities[30]. The positive effect of country area on establishment richness for birds and amphibians suggests that unaided dispersal pathways may contribute to establishment richness in traded alien species for both taxa. Larger areas are likely to have longer borders with neighboring countries, which increases the chance of invasions from other countries by unaided dispersal pathways of established species[27,42]. The positive effect of population density on establishment richness for birds and amphibians may be because most exotic pets are released in densely populated areas[43,44], which result in high colonization pressures.

The alien species that are traded in a greater number of countries or over larger areas often have larger trade volumes, which has previously been linked to a positive association with establishment success[26,27]. Species traded over larger areas are also more likely to encounter suitable environments, promoting population establishment of released or escaped pets[15]. Additionally, establishment success for traded alien species can also be influenced by various species traits, such as morphological characteristics, reproductive traits or habitat requirements[45–47]. Market factors such as price and availability of species[48,49], desire for alien pets[50], and other economic uses of species in trade[12] may affect the demand and subsequent invasion success of traded species. Climate change can also affect the suitability of ecosystems for alien species, thereby facilitating their establishment in new areas[51]. Understanding the effects of these factors on the establishment of alien species in trade at a global scale is an important area for further research.

Overall, our findings identify the significant challenges to global biosecurity posed by wildlife trade. The large number of alien species present in global trade represent high colonization pressures for alien vertebrate species to establish populations in novel areas. With the increasing globalization (e.g. trade in more countries or areas) of the exotic pet trade, the establishment likelihood of alien vertebrate species will rise. Countries with a high number of alien species in trade, such as the United States, Western European countries, Canada, and island nations are considered current and future hotspots for invasions. Countries with rapidly increasing GDPpc or trading power, such as developed or emerging countries in South and East Asia, are likely to be future hotspots for alien vertebrate establishment. As economies develop and international trade expands, these countries may see a rise in the volume and diversity of traded species, which can elevate the risk of alien species introduction and establishment in their respective regions. It is important for these countries to proactively respond to these challenges by prioritizing effective biosecurity measures, strengthening regulations on wildlife trade, and implementing robust monitoring and risk assessment systems to mitigate the unintended introduction of invasive species and the associated future environmental, economic and zoonotic disease impacts they pose[52].

The COVID-19 outbreak has stimulated calls for a global wildlife trade ban, but such actions may negatively affect the livelihoods of people depending on wildlife and fuel illegal wildlife trade[24,53]. Reducing the likelihood of release or escape of exotic pets is key to managing the invasion risks posed by wildlife trade. Governments need to draft policies that effectively reduce the potential for such release or escape. Few

countries are yet to implement sufficient regulations or legislations to monitor and manage the release or escape of exotic pets[54].

Due to the difficulty of eradicating invasive species once they become established in previously unoccupied areas, the highest priority for each country should be to develop national capacities for early detection, monitoring, and rapid response to introduced species incursions[55,56]. To accomplish this, it is critical to utilize both existing and emerging technologies for the early detection and monitoring of introduced species, such as environmental DNA, remote sensing, chemical ecology, and internet-based applications that engage citizen scientists[55]. More efforts should be directed towards advancing technological capacities for the rapid detection, identification, reporting and response to introduced species.

## Methods

### Global live vertebrate trade database (GLVTD)

**Extracting data on live wildlife trade from databases.** We extracted data on the trade in live wildlife for terrestrial vertebrates from the CITES Trade Database, Law Enforcement Management Information System (LEMIS) and International Species Information System (ISIS). The CITES Trade Database (https://trade.cites.org/, last visited on 1 August 2022) is developed and maintained by the United Nations Environment Programme World Conservation Monitoring Centre on behalf of the CITES Secretariat. This database includes records of international trade in CITES–listed species reported by CITES Parties annually. While the CITES Trade Database covers data on legal trade by member nations (Parties), some Parties may also report seizure events under "Source I" in the database[57]. LEMIS is based on The United States Fish and Wildlife Service's (USFWS) Law Enforcement Management Information System data derived from legally mandated reports submitted to USFWS (https://www.fws.gov/library/collections/office-law-enforcement-importexport-data, last visited on 30 April 2023), containing records on US imports and exports of both live wildlife and wildlife derivatives. Like other studies[13], we treated each transaction as a trade record, and obtained data on all transactions for mammals, birds, reptiles and amphibians from the CITES Trade Database (version 2022.1, 48 files containing 23,680,557 records) between 1975 to 2021 and LEMIS between 1999–2020 (56 files containing 5,944,959 records). We curated, cleaned, and compiled data from CITES Trade Database following recommendations by Challender and colleagues[57], and LEMIS following Watters and colleagues[58]. We first collected data by choosing "live" in the column of "term" in CITES or "LIV" in the column of "Wildlf Desc" in LEMIS for each taxon. Here,"live" or "LIV" indicates live specimens, covering different units used in the column of "Unit"for CITES and LEMIS, such as NO (number of individuals),weight (grams, kilograms, liters), boxes, shipments, and others. We then obtained data on scientific name, class, and family of species under "live" or "LIV" filtering, importer or, exporter, and year for each transaction. Live wildlife in CITES and LEMIS is traded for a variety of purposes, including personal use (e.g. pets), commercial sale, medicinal and scientific purposes, use by law enforcement, and education, conservation, hunting and display (e.g., zoos, breeding in captivity, reintroduction or introduction in the wild and circus or travelling exhibitions). As we are focused on species richness in trade, we did not consider the trade volume ("Quantity" in CITES and "Qty" in LEMIS) and source of species. We also did not include "eggs" (live) trade from CITES and LEMIS because species in live specimen trade generally covered those species involved in eggs trade. We performed quality control of data by excluding records with duplicated lines or the same importer and exporter countries, those with no scientific name, and unidentified or hybrid species.

ISIS is a network of 837 zoos and aquaria that shares information about 2.5 million animals of more than 10,000 species among member institutions[59]. The ISIS Database (ISIS.IUCN.Matching.xls, containing 94,877 records of mammals, birds, reptiles, and amphibians) compiled by Conde and colleagues[59] holds the most comprehensive information on animals kept in the zoos across the world in 2011. We collected data on scientific name, class and family of animals kept in zoos and countries or regions from ISIS Databases for each taxon. Not all animals in zoos are sourced from trade or for trade, and ISIS Database does not provide information on the source of animals in zoos, and records of transactions among zoos or intuitions. It is difficult to identify which species or which zoos were involved in trade. Inclusion of all species in ISIS Database as traded ones would overestimate number of species traded for zoos. Conde and colleagues[59] suggested that threatened species (categorized as vulnerable, endangered, or critically endangered) are kept or bred in zoos mainly for conservation purposes (not for commerce), such as ex-situ conservation, reintroduction programs for population persistence or conservation campaign. We therefore excluded data on threatened species from ISIS (635 species, Table S10), assuming that they were not involved in trade. This exclusion might result in a conservative estimation of species involved in trade for zoos, but would not have an effect on threatened species that were recorded in other databases (CITES, LEMIS and OTAPS). Approximately 80% (508/635 species) of the threatened species in ISIS (Table S10) have trade records in other databases (CITES, LEMIS and OTAPS). The number of threatened species from ISIS that are not shared with other databases accounts only for 1.63% (127/7,780 species) of total species in our database (ranged from 3.85% (48/1247 species) of mammals, 1.07% (37/3,451 species) of birds, 0.97% (22/2278 species) to 2.49 % (20/804 species) of amphibians) (Table S10), indicating little effect of the exclusion on total number of species in GLVTD.

**Data on contemporary online trade of wildlife.** We systematically searched websites offering live wildlife trade for pets (online pet shops), public display (zoos) and for other uses such as food[60–62]. We crawled data on listings (advertisements and posts) from these websites. We built keys for species names and extracted information on species names and countries from the crawled data.

**Searching for the websites of live wildlife trade.** We searched for websites involved in live wildlife trade on Google Hong Kong (http://www.google.com.hk) for each of 193 countries from March to May 2022. We performed searches for each country using search phases, in English, such as "taxon (each group of mammals, birds, reptiles and amphibians) + for sale + country name". We additionally searched websites in other languages (up to three) for each country using Google Translation, based on widely spoken languages (official or national languages) (quickgs.com). In total, we used 1414 phases in 69 languages for the searches (Supplementary Data 5–8). To determine a cutoff point that balanced the quality of search results with search effort we randomly selected 10 countries in Europe and Asia and browsed each website using the URLs returned by the search phases (in English) to choose[62]. This browsing process revealed that when 20 consecutive websites in a list of returned URLs did not show listings (advertisements or posts) of exotic pets or live wildlife, additional browsing was unlikely to find other relevant websites in the rest of the list. We therefore applied this cutoff point to all our searches. We manually browsed websites, with two persons (YL and ZXL) initially performing the website search together to establish consistent practices, then by browsing separately[61]. We browsed 95,965 websites across 193 countries in total and identified 1463 websites involved in live wildlife trade across 177 countries. These websites used 47 languages, with approximately 55% (799) being in English (Table S11), while 44% used other languages, and 1% were a mix of two languages.

**Scraping online data.** We used the Web Scraper tool on the Chrome browser (https://www.webscraper.io/) between Jun-August 2022 to scrape and extract data on title, contents, scientific name and price

of pets, locality (city in a country), date of listings posted, and URLs, for all pages stored within a website during the search timeframe. The Web Scraper is a web scraping tool with many advanced features to get exact information from websites. It can perform data scraping from multiple pages, multiple data extraction types (text, images, URLs, and more), scraping data from dynamic pages (JavaScript + AJAX, infinite scroll), browsing scraped data and other functions. We created a sitemap for each website to be crawled and pasted the URL root (webpage 1) of a website for this sitemap in Start URL. We then created a loop through the web pages by repeatedly going to the next page for the scraper by establishing a new column for this function. We clicked on 'Add new selector'; under root window, we input a name for the column in ID box, selecting 'Pagination (Beta)' in the Type box. We clicked on 'Select' in the Selectors box and then on Paging button (Next or 2) in the webpage. We selected both root and name of this column in the Parents selector box, and saved these settings by finishing pagination settings. We gave a name for the column of listings and selected 'element' in the Type box (for websites with scrolling listings, selected 'Element Scroll Down'), and clicked on 'Select' in the Selectors box and then on two listings in the webpage (the scraper could automatically select others with same structure). We checked if all listings were selected (in red) by clicking on the Element preview button. If any listings were not selected due to variations in their structure, we manually clicked on those listings. We then saved the settings by finishing the selection of listings.

Following these configuration steps, we performed the data scraping as follows:

Cycle. For websites with pages of listings containing all data to be crawled, we simply input a name of a phase to be crawled in ID box, selected 'text' in the Type box, clicked on 'Select' in the Selectors box, and selected the phase in a listing in the webpage, and saved the settings.

Crawls. For websites with pages (cycle or not) showing parts of information and other information contained in different levels of subordinate linked pages, we selected a name for the phase linked to the information in ID, then selected 'link" in the Type box and selected the phase in the webpage. The name for this phrase will show in the Parents box. In root window, we clicked on the name, which showed the linked page in the webpage, and we set new name in ID and selected a phase to be crawled. For the deeper links in a website, we used the procedure as above.

We clicked on the sitemap file in the Toolbar after all settings finished, and then on "Scrape" to open a configuration table, then on 'Start Scraping' by default setting (Request interval and Page load delay (2000 ms)) to run the program. We downloaded the sitemap in XLSX file once the program was done. We crawled all websites relating to wildlife trade, except for one website that displayed its listings in PDF format. In this case, we directly downloaded the PDF file and copied information from the PDF file into the text.

After completing the web scraping process, we checked the consistency between the crawled data and the listings on each webpage contained in a website. If we found any listings missing from the crawled data, we made necessary adjustments to the settings of the scraper and re-scraped the data from the website. In some cases, we manually transcribed the missed information to save time if only several listings on a website were lacking from the crawled data. To maintain consistent scraping protocols, the authors participated in a training course provided by Web Scarper (seven authors: Z.X.L., Y.Y.L., J.C.D., M.L.N., J.Y.Z., J.Z. and Y.L.). Following the training, each author then independently conducted the crawling and scraping process.

**Data on keys.** We gathered keys from different databases. We obtained data on scientific names, synonyms, and common names in

different languages for mammals, birds, reptiles, and amphibians from the IUCN Red List by Web Scraper, and downloaded data from relevant taxonomic websites (mammaldiversity.org; avibase.bsc-eoc.org/; reptile-database.reptarium.cz/; amphibiaweb.org/, last visited on 17 Sep. 2022) (Excel files). We also obtained trade names of species in English, French and Spanish from the CITES Trade Database (2022 V.1), and specific names of species in English from LEMIS. In total, we obtained 484,470 species names, including 47,041 names for mammals, 304,246 names for birds, 93,401 names for reptiles and 39,782 names for amphibians.

**Extracting species names from crawled data.** We extracted string keys for species names from titles, contents or scientific names in the data crawled using the formula of Lookup function combined Find function in Excel 2016 as follows[63]:

$$Formula = LOOKUP(1,0/FIND(X\$i:X\$j, Yi), X\$i:X\$j) \qquad (1)$$

Where X is the column of the keys that we wanted to look up, with i and j indicating the range where keys were located in rows. The column X was sorted in ascending order based on the number of characters contained in a string using the Len function. Y identifies the columns including data crawled (titles, contents or scientific names) where we searched for keys. As the Find function is case sensitive, we transformed keys and crawled data (titles, contents or scientific names) to lowercase using the Lower function before extraction. We matched the extracted keys with the scientific names in the key database using the VLOOKUP function:

$$Formula = VLOOKUP(xi, y:z, 2, 0) \qquad (2)$$

Where X is the column of the extracted keys, Y is the columns containing synonyms, common names, traded names, or specific names, and z is the column with corresponding scientific names.

**Publications on historical online trade and physical markets.** We searched on Google Scholar for publications using the search phases "taxon (each group of mammals, birds, reptiles and amphibians) name +for sale+country" in English for each of 193 nations (Supplementary Data 5–8). We browsed each of publications returned, reviewed its title and abstract, and excluded studies solely on data from the CITES Trade Database, LEMIS and ISIS. We stopped searching for publications if 20 consecutive publications in a list of returned URLs did not contain studies on exotic pets or live wildlife. We downloaded 110 publications in total (Supplementary Data 9), including studies on online trade, physical stores or markets, zoos, those on both online trade and physical markets, and on databases of wildlife trade[19]. Because we focused on the list of alien species involved in live wildlife trade, we put publications on legal or illegal wildlife trade, or both together for analysis. We extracted records of the species names and countries involved in live wildlife trade from these publications.

**Identifying alien species in GLVTD**

We combined datasets from CITES, LEMIS, ISIS, contemporary online trade and publications on historical online trade and physical stores (shortened as online trade and physical store, OTAPS) into a list of species traded in countries or regions. Different taxonomies were used in different data sources, which would inflate the list of species in trade and bias the delimiting of native ranges for some species. We resolved species names and higher-level taxa according to the taxonomy of the IUCN Red List. We aligned the list of traded species with those of scientific names and synonyms in the IUCN Red List using the VLOOKUP function in Excel. We obtained a final list of matched species in trade (trade data) by removing duplications. This list includes 6136 species collected from CITES, EMIS and ISIS, 3204 species from

contemporary online trade, and 3551 species from publications on historical online trade and physical markets. Unmatched names might be due to typing errors, unaccepted names, or different taxonomies used, and were excluded from downstream analysis.

We obtained data on the geographic ranges (based on global administrative areas at country or region level) of species from the IUCN Red List. We defined the native range of a species as the countries or regions that have native extant or native possibly extant presence of the species or those with extinct or possible extinct range of the species. We downloaded spatial data on geographic range for each taxon (https://www.iucnredlist.org/resources/spatial-data-download, last visited on 30 Nov. 2022) and Database of Global Administrative areas (GADM, version 2.8) (GADM.org). We derived data on countries or regions where a species occurs by overlapping the geographical map of a species with GADM using ArcMap. The maps of some species could not be categorized into specific countries or regions due to overlapping occurrences in marine habitats or on tiny islets. We obtained data on the country or region level native range for these species by visiting the website of each species on IUCN Red List and downloaded data on their geographical ranges (14 species having no such data, Table S1). We obtained the list of native countries or regions in which a species naturally occurs by excluding species without range data and countries with extant introduced presence of species.

We matched all combinations (a traded species name + a country or region name as a combination) of a traded species and each of countries or regions in trade with those combinations of the species and a native country or region (a species name + a native country or region name) using the VLOOKUP function (2). Here, X is the column for combinations of traded species and countries or regions in trade, and Y is the column containing combinations of species and native countries or regions, and Z is the column with the string "True". While a matched combination for a traded species (showing "True") indicates that the species was traded in a native country or region, an unmatched one ("#N/A") suggest that it was traded outside its native range, namely an alien species in trade. We transformed unmatched or matched combinations to columns and counted alien richness across countries or regions.

## Data on established alien species

We obtained data on established alien terrestrial vertebrate species and their distributions (established countries) from a number of databases (the Global Invasive Species Database (GISD, http://www.iucngisd.org/gisd/, last visited on 30 May 2021; mammals[64–66], birds[67], reptiles and amphibians[40,68–70])). We collected additional data by retrieving information on the geographical ranges of species from the IUCN Red List and including the species that have an extant introduced presence in countries. We also reviewed each paper published in the journal *BioInvasions Records* between 2015 and 2022 and extracted records (species and distribution) of established vertebrates (Supplementary Data 10). We checked the species names of established vertebrates against the scientific names and synonyms in the IUCN Red List and excluded repeated names from our list. We included a total of 1041 established vertebrate species (Supplementary Data 3). We matched the list of established vertebrates with trade data, and identified established alien species by trade as those that were involved in the live wildlife trade. We mapped the richness of alien species in trade and established alien species richness in ArcGIS.

## Data on socio-economic and environmental factors across countries

We obtained data on area, GDP, population size and total value of import and export goods (e.g., commercial trade amount) for each country in 2015 from the World Bank (http://data.worldbank.org/, last visited 15 April 2023). GDPpc was calculated as GDP divided by human

population size, and population density as population size divided by area. We identified a country as an island nation (e.g. insularity) based on world atlas (https://www.worldatlas.com/geography/island-countries-of-the-world.html, last visited on 15 June 2022). The income categories of nations were identified according to the analytical categories of World Bank based on Gross National Income per capita (GNI per capita) US$ in 2015 (https://data.worldbank.org/indicator/, last visited on 24 Dec. 2022). Data on annual mean temperature and precipitation were calculated from the spatial data set for the period 1950 to 2000 at a resolution of 10 arc minutes from WorldClim (www.worldclim.org). We used data from Moura and Jetz[71] on the proportion of undiscovered vertebrate species in each country as a metric of sampling effort. We obtained data on the congeneric richness of each taxonomic group from each country from the IUCN Red List (https://www.iucnredlist.org/search, last visited on 15 July 2021).

## Statistical analysis

We identified the geographical patterns of alien species number in relation to human population size, GDPpc and commercial trade amount for each taxon for countries categorized as having high or upper middle income using univariate linear-mixed models, where the number of alien species was the response variable, and each socioeconomic factor was included as a predictor. The biogeographical realms in which a country is located (the midpoint of its latitudinal and longitudinal ranges) was included as a random variable to account for geographic autocorrelation. We categorized biogeographical realms following the definition of Olson and colleagues[72]: Afrotropics (including Madagascar), Australasia, Indo-Malay, Nearctic, Neotropics, Palaearctic and Oceania. Human population size, GDPpc and commercial trade amount were log transformed and the number of alien species was log (x + 1) transformed to improve their linearity before analysis.

We compared differences in the number of countries or areas involved in trade between established and unestablished alien species using univariate generalized linear mixed models (GLMMs) with a logit link function and a binomial error distribution, with the establishment of species (established=1, unestablished=0) as the response variable and the number of countries or areas involved in trade as a predictor across alien species for each taxon. To account for taxonomic autocorrelation, we included order/family/genus as nested random variables in the model.

We identified the effects of predictors on establishment richness of traded alien vertebrates across countries for each taxonomic group separately, using multimodel inference. The full model was a linear mixed model (LMM) with established alien species richness (establishment richness) as the response variable, and nine factors as predictors (fixed effects: area, population density, GDPpc, colonization pressure, insularity (binary variable, island country or not)), annual mean temperature, annual mean precipitation, congeneric richness and sampling effort (proportion of undiscovered species). To account for geographical autocorrelation, we included biogeographical realm as a random variable in the model. Area, population density, GDPpc and mean precipitation were log transformed, and establishment richness of alien species, number of alien species in trade, and congeneric richness were log (1+x) transformed to improve their linearity. We constructed 512 models ($2^9$) representing all combinations of the predictor variables. We calculated standardized estimates for regression coefficients and standard errors for each variable[35]. We calculated the statistical significance of the coefficient for each predictor based on a z-score with a 95% upper confidence limit ($|z| \geq 1.96$).

We also performed model selection by ranking the performance of models based on the Akaike information criterion adjusted for small samples (AICc)[73]. We identified those models that were within 2 AICc units of the highest-ranked models (i.e., $\Delta$AICc $\leq 2$) as top models.

We performed network analysis to quantify the global flows of traded alien species and traded alien species with established populations (established aliens) from their native and alien countries[40,74]. Following Sander and colleagues[75], we classified the world into 8 economic regions: South and East Asia, Mideast and Central Asia, Africa, Europe, North America, Central America, South America and Oceania. We identified major donor and recipient regions in terms of number of species.

We performed GLMMs using the 'glmmTMB' function in the TMB and glmmTMB packages. We conducted LMMs using the 'lmer' function in the lme4 package. We ran the model-averaging analysis using 'dredge' and 'model.avg' in the MuMIn package. We carried out network analysis using the Circlize package based on the procedures of Sander and colleagues[75]. These analyses were conducted in R Studio 2022 (https://github.com/rstudio/rstudio). The R scripts used in this study are provided in Supplementary Code 1.zip.

### Reporting summary
Further information on research design is available in the Nature Portfolio Reporting Summary linked to this article.

## Data availability
The GLVTD database, which contains identifiable information of commercial websites, is available from the corresponding author (YL, liym@ioz.ac.cn) on request. A reply to a data access request will be provided within one week from the date of the request. Other data from this study can be found in Supplementary Data 1–10 or in Figshare: https://doi.org/10.6084/m9.figshare.23291966. Source data are provided with this paper.

## Code availability
R scripts are provided in Supplementary Code 1.

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

## Acknowledgements

We thank Dr. X Liu and two colleagues for their comments on the draft. We are grateful to Jiaqi Zhang for his help for reorganizing the figures. YL is supported by grants from National Natural Science Foundation of China (32030070), Second Tibetan Plateau Scientific Expedition and Research (STEP) Program (2019QZKK0501), High-Level Talents Research Start-Up Project of Hebei University (050001-521100222045), Hebei Natural Science Foundation (C2022201042), and China's Biodiversity Observation Network (Sino-BON).

## Author contributions

Conceptualization: Y.L. Methodology: Y.L., Z.X.L. Investigation: Y.L., Z.X.L., T.S., T.D., W.L., F.W., Z.H.L., Y.Y.L., J.D., M.N., J.Y.Z., J.Z., J.Y., S.W. Visualization: Y.L., Z.X.L., W.L., Y.Y.L. Funding acquisition, project administration, supervision, writing – original draft: YL. Writing – review & editing: T.M.B., Y.L., F.W.

## Competing interests

The authors declare no competing interests.
