## [Peer Review File · Nature Communications]

Quantifying Global Colonization Pressures of Alien Vertebrates from Wildlife TradeREVIEWER COMMENTS

Reviewer #1 (Remarks to the Author):

Overall, this is a strong study and write-up that presents important, timely findings relevant for the moment (following calls for action related to increased surveillance of wildlife trade following COVID-19). The study quantifies invasive species associated with (and possibly resulting from) the wildlife trade. I am not aware of any similar studies at a global scale, and I can envision that this will be a bedrock paper that serves as evidence to motivate management and policy. Impressive data collection and synthesis. Well written: clear language, poignant framing, organized, compelling storytelling, good figures. Analyses are simple yet appropriate and revealing of the major take-aways.

Most of my comments offer ideas for how the authors could enhance their study from a strong paper to a stellar, crystal clear paper. A few over-arching suggestions:

- At a minimum, the authors should add the disclaimer that some of these colonizations may result from processes other than wildlife trade. There are many reasons how and why a species could establish outside its native range. If the authors have the word count to take it further, I'd be interested in seeing their thoughts on how we might know (maybe as a suggestion for future research) if a species colonized as a result of wildlife trade vs some other reason.
- The authors need to explain several of their results (see detailed comments) in greater detail to more fully interpret and explain hypotheses for the patterns.
- The conclusion needs to be fleshed out more and linked to policy and specific examples of what's working and what's not to address this challenge currently. Right now, as a policymaker, I'm not sure how to apply the findings; the authors are on the cusp of making these connections explicit and applicable for readers.
- I'd be interested to know which species in the dataset have NOT colonized as a result of trade? Is there something unique or different about these compared to those species that

have colonized? Do you expect that they will colonize in the future, or are their characteristics and ecological needs less conducive to colonization?

- Given the authors' bilingual abilities (e.g. Fig 4), I wonder if it could be beneficial to include a non-English abstract and figures (and other key summary aspects) in the paper and/or Supplemental materials. This could increase accessibility for readers whose first language is not English (and help promote the paper where the authors are based).

Detailed comments:

Please double check that all acronyms are defined at first mention in the text (e.g., CITES and LEMIS on line 97, ISIS on line 98).

Line 105 – It could be interesting to know which species were excluded, because it highlights an important gap. This is optional and low priority; I'd suggest including this list in the supplemental materials.

Line 117 – Please include the results for each country in the supp materials so readers don't only have to rely on the figure (hard to see the result for small countries).

Line 120-122 – It's fine to report the general pattern, and it could additionally be interesting to know if this is statistically correlated for all species (how strong the pattern is; as the authors have done by taxa). That income and area would correlate with the number of alien species traded is perhaps not surprising but a fascinating and important finding nonetheless.

Line 126-133 – Very interesting. It might also be relevant here to compare to a factor related to trade and the economy (a possible indicator of commercial trade amounts generally; I know you also mention this result on line 158 for a different pattern). I see the hypothesis to explain this pattern in lines 192-196. Is there more you can say here? I am curious if there may be other possible reasons why this pattern emerges.

Line 148 – If South Sudan was the only country where there was no establishment, please

make this more explicit (otherwise it leaves me wondering why South Sudan was mentioned).

Line 197-202 – Can you say more? Would like to see this explored in greater depth. Please explain what colonization pressure means in terms of the realities of wildlife trade once an individual is brought into a country.

Line 203-213 – Several sentences in this paragraph need to be simplified so readers less familiar with the ecological terms can understand, including: “Consistently positive influences of insularity on establishment richness across taxonomic groups” and “...effects on lineage survival probability of reduced biodiversity or increased ecological naiveté of native insular communities”. I appreciate that in this paragraph you link and explain the results to the aspects of the trade itself – please explain even further (same for above in 197-202).

Line 221-224 – This recommendation follows on the heels of numerous calls for increased disease testing following COVID-19. I’d recommend citing a few papers to support your statement and connect to this larger conversation.

Line 214-228: All this is true, however I urge the authors to consider spending your precious word count on the most impactful recommendations for decision-makers. Everyone who works on wildlife trade already knows that the CITES Trade Database has shortfalls and needs to be improved (this has been written about countless times), so I’d recommend replacing line 224-225 with a more unique and important recommendation. The timing of this study’s publication falls on the heels of the major findings and calls for action following COVID-19 around the lack of monitoring and testing of wildlife trade generally (including for disease). Can you build on those calls to action to include monitoring alien species and advancing more rigorous policy, regulation and law related to that? For example, the United State’s Lacey Act has powerful regulations of injurious species that enable species to be listed to prevent colonizations from trade (for example) – is this an example you may want to highlight (or examples from other countries) for other countries to emulate, or that need to be taken further? Would be great to see some specific linkages to policy or higher-level

decisions beyond general calls for more data.

Table 1 – What is “biological realm”? I don’t see this term elsewhere in the paper and am not sure what factors it includes.

Fig. 1 – Please define all acronyms that are shown in the figure within the caption text.

Fig. 2 – Please briefly explain the four categories in the caption, so it makes sense to readers without having to refer to the methods. The blue category especially could benefit from a definition. For example, which categories are subsets of other categories (e.g., do yellow and green categories add up to the total in red?).

Fig 3 – I would encourage the authors/edits to print this image as large as possible in the published layout, so readers can make out the different countries.

Fig 4 – Please convert axis number/name labels to English.

Fig 5 – The regional labels are so spaced out that they are challenging to read. Please label each color individually rather than as a region (e.g., what region are the pale yellow and green?). I am shocked to see the proportional trade sourced from Asia.

Reviewer #2 (Remarks to the Author):

Please see the attached track changes for comments.

As currently written, the manuscript lacks significant methodological specificity and cannot be reproduced. Can the database be made public?

Reviewer #3 (Remarks to the Author):

Overall, this paper is bringing together broader data on different types of animal invasives and WT, where before we mainly had smaller regional or taxonomic studies. In doing that it is addressing a research gap. However, the methods are a bit vague in places, and there are

a lot of different data sources used, which are all hard to use even on their own, so I would need some more detail of the methods for things like the online sampling, literature search, and CITES analysis. The results are interesting but the discussion is also vague in some key places, and would benefit from more grounding in wider wildlife trade literature, better linking of conclusions to the findings, and more tangible recommendations for applications of these findings (e.g. rather than saying 'increase surveillance', give some actual recommendations for how to address this).

Specific comments

L98 - I don't know what ISIS is

L101 - after all of this it is hard to know what 'these groups' means - I worked it out eventually but clarification would be helpful

L121 - as written this sounds like Western Europe is a country

L174 how does this reflect increasing complexities - this exchange pattern has been the case for some time and inter-continental trade is not an emerging issue

L193 I don't think you have provided enough to support this argument about range size - it reads like an add on that doesn't really link to your results

L221 - what does 'surveillance' mean in this context? It would be useful to have a concrete recommendation here if you are saying it is urgent, as this is not clear.

L224 - not clear what you mean here about the CITES Trade Database - it is not designed to monitor alien species, especially as many potential invasives aren't listed on CITES, which you note next, but then why would it ever be expected to be sufficient for this? Surely Customs data would be more applicable?

L228 - the end of the paper doesn't really give clear ideas of what to do - it sounds a bit vague and I think this is a shame because the data have the potential to underpin some strong policy recommendations.

L236 the CITES TD is managed by UNEP not UNDP

L239 the CITES TD is a legal trade database - some Parties might report Source I as a seizure event but it is not that accurate to say it covers seizure data - see here for ways to ensure trade is not mischaracterized using these data

<https://conbio.onlinelibrary.wiley.com/doi/full/10.1111/conl.12832>

L248 what purpose codes and other search terms did you use (your stated search of 'Live'

on the CITES data would return every taxa listed as well as animals, including plants and taxa not in commercial trade).

L262 The methods for the online trade sampling are not clear enough - does 'browse' mean somebody physically visited 95,000 websites?

L345 what does 'intensively' searched mean? How did you select keywords? Did you follow any systematic online search protocols e.g.

<https://www.cambridge.org/core/journals/oryx/article/systematic-survey-of-online-trade-trade-in-saiga-antelope-horn-on-russianlanguage-websites/3422BF90210A106E383E199D0E5D56CE> - if not please note how you ensured that the sampling was picking up as much as possible.

Figure 1 - this looks really nice but is very hard to interpret (these multiple Venn diagrams always are!)

Figure 2 - the caption needs more detail

The maps - The silhouettes are a bit confusing where they are

Point-by-Point Responses to Reviewer Comments

Reviewer #1 (Remarks to the Author):

Overall, this is a strong study and write-up that presents important, timely findings relevant for the moment (following calls for action related to increased surveillance of wildlife trade following COVID-19). The study quantifies invasive species associated with (and possibly resulting from) the wildlife trade. I am not aware of any similar studies at a global scale, and I can envision that this will be a bedrock paper that serves as evidence to motivate management and policy. Impressive data collection and synthesis. Well written: clear language, poignant framing, organized, compelling storytelling, good figures. Analyses are simple yet appropriate and revealing of the major take-aways.

Our Reply: Thank you for your overall positive comments on our manuscript. We have considered your comments carefully, and made the changes following your comments. We also corrected errors in LEMIS data filtering in the previous version of the manuscript, which result in an overestimate of some alien richness values (mainly the USA and global).

Most of my comments offer ideas for how the authors could enhance their study from a strong paper to a stellar, crystal clear paper. A few over-arching suggestions:

- At a minimum, the authors should add the disclaimer that some of these colonizations may result from processes other than wildlife trade. There are many reasons how and why a species could establish outside its native range. If the authors have the word count to take it further, I'd be interested in seeing their thoughts on how we might know (maybe as a suggestion for future research) if a species colonized as a result of wildlife trade vs some other reason.

Our reply: The suggestion is very good, and we appreciate it very much. We have added three sentences to note that some colonization may not result from wildlife trade. We did this by discussing the area effect on establishment richness for birds and amphibians based on results from multimodel inference and model selection, which may be due to the unaided dispersal pathway rather than wildlife trade pathway, and explained the potential relationships between this area effect, country area and border length.

- The authors need to explain several of their results (see detailed comments) in greater detail to more fully interpret and explain hypotheses for the patterns.

Our reply: Good points, which improve the quality of our work. Fig. 3 originally looked as if socio-economic factors drive the patterns of alien species number, and we only described this without performing deep analysis in the previous version. Now we have re-collected data on commercial trade amount (total value of import and export goods) as suggested, and added new analyses (linear mixed models, LMMs) on relationships between alien species number and socio-economic factors (GDPpc, population and commercial trade amount) explicitly to show the patterns of alien species. Yes, all three factors have significant effects on alien species number. We have also added a new Table (Table S3) to show the results of LMMs, and added some sentences to describe the results and some discussions on the patterns in the Discussion.

- The conclusion needs to be fleshed out more and linked to policy and specific examples of what's working and what's not to address this challenge currently. Right now, as a policymaker, I'm not sure how to apply the findings; the authors are on the cusp of making these connections explicit and applicable for readers.

Our reply: We have re-written the whole conclusions by linking our findings to applications, and giving more tangible recommendations for the applications from identifying current and future invasion hotspots to drafting regulations on release or escaping of exotic pets and early detection and rapid responses. We have also additionally cited several references for the reviewing wildlife trade.

- I'd be interested to know which species in the dataset have NOT colonized as a result of trade? Is there something unique or different about these compared to those species that have colonized? Do you expect that they will colonize in the future, or are their characteristics and ecological needs less conducive to colonization?

Our reply: Good suggestions. We have added new analyses of comparisons on differences in characteristics (number of countries involved in trade, and associated areas) between establish and

unestablished species for each taxon using generalized linear mixed models. We added a new table (Table S5) and two figs (Fig S4 and S5) to show the results, and added discussions about what might be expected to colonize in future.

- Given the authors' bilingual abilities (e.g. Fig 4), I wonder if it could be beneficial to include a non-English abstract and figures (and other key summary aspects) in the paper and/or Supplemental materials. This could increase accessibility for readers whose first language is not English (and help promote the paper where the authors are based).

Our reply: A good suggestion. Now we have added Abstract in Chinese in supplementary materials.

Detailed comments:

Please double check that all acronyms are defined at first mention in the text (e.g., CITES and LEMIS on line 97, ISIS on line 98).

Our reply: We have added whole phases for CITES, LEMIS and ISIS here.

Line 105 – It could be interesting to know which species were excluded, because it highlights an important gap. This is optional and low priority; I'd suggest including this list in the supplemental materials.

Our reply: We originally aligned the traded countries and regions with geographical regions based on spatial data of IUCN Red List. There were more than 100 species without geographical range or unmatched spatial data with World Administration Areas exactly. We resolved this question by recollecting the data on geographical range by visiting website of each of such species (spatial data and data on website are highly consistent), and realigned all data, and re-performed all analyses based the new alignment. Now there are 14 species without geographical range data totally. We have added Table S1 showing species without geographical range, and added more words to describe how we obtain the geographical range in more details in Methods.

Line 117 – Please include the results for each country in the supp materials so readers don't only have to rely on the figure (hard to see the result for small countries).

Our reply: Thanks for the suggestion. We have added a supplementary data 2 and 4 showing the number of alien species and established species for each taxon across the countries and regions, which are consistent with Fig 3 and Fig 5.

Line 120-122 – It's fine to report the general pattern, and it could additionally be interesting to know if this is statistically correlated for all species (how strong the pattern is; as the authors have done by taxa). That income and area would correlate with the number of alien species traded is perhaps not surprising but a fascinating and important finding nonetheless.

Our reply: Please see our reply to the general comment 2.

Line 126-133 – Very interesting. It might also be relevant here to compare to a factor related to trade and the economy (a possible indicator of commercial trade amounts generally; I know you also mention this result on line 158 for a different pattern). I see the hypothesis to explain this pattern in lines 192-196. Is there more you can say here? I am curious if there may be other possible reasons why this pattern emerges.

Our reply: Please see our reply to the general comment 2.

Line 148 – If South Sudan was the only country where there was no establishment, please make this more explicit (otherwise it leaves me wondering why South Sudan was mentioned).

Our reply: We have deleted this sentence.

Line 197-202 – Can you say more? Would like to see this explored in greater depth. Please explain what colonization pressure means in terms of the realities of wildlife trade once an individual is brought into a country.

Our reply: We have added words to clarify the number of traded alien species as a measure of colonization pressure in a country in Results, and added more sentence here to discuss the colonization pressure.

Line 203-213 – Several sentences in this paragraph need to be simplified so readers less familiar with the ecological terms can understand, including: “Consistently positive influences of insularity on establishment richness across taxonomic groups” and “...effects on lineage survival probability of reduced biodiversity or increased ecological naiveté of native insular communities”. I appreciate that in this paragraph you link and explain the results to the aspects of the trade itself – please explain even further (same for above in 197-202).

Our reply: We have modified these two paragraphs to be concise and clarify what we are saying here.

Line 221-224 – This recommendation follows on the heels of numerous calls for increased disease testing following COVID-19. I’d recommend citing a few papers to support your statement and connect to this larger conversation.

Line 214-228: All this is true, however I urge the authors to consider spending your precious word count on the most impactful recommendations for decision-makers. Everyone who works on wildlife trade already knows that the CITES Trade Database has shortfalls and needs to be improved (this has been written about countless times), so I’d recommend replacing line 224-225 with a more unique and important recommendation. The timing of this study’s publication falls on the heels of the major findings and calls for action following COVID-19 around the lack of monitoring and testing of wildlife trade generally (including for disease). Can you build on those calls to action to include monitoring alien species and advancing more rigorous policy, regulation and law related to that? For example, the United State’s Lacey Act has powerful regulations of injurious species that enable species to be listed to prevent colonizations from trade (for example) – is this an example you may want to highlight (or examples from other countries) for other countries to emulate, or that need to be taken further? Would be great to see some specific linkages to policy or higher-level decisions beyond general calls for more data.

Our reply: We have re-written the whole conclusions by linking our findings to applications, and

giving more tangible recommendations for the applications from identifying current and future invasion hotspots to drafting regulations on release or escaping of exotic pets and early detection and rapid responses. We have also additionally cited several references for the reviewing wildlife trade.

Table 1 – What is “biological realm”? I don’t see this term elsewhere in the paper and am not sure what factors it includes.

Our reply: sorry for the mistake. We have corrected this phase.

Fig. 1 – Please define all acronyms that are shown in the figure within the caption text.

Our reply: We have added whole phrases to all acronyms in the legend.

Fig. 2 – Please briefly explain the four categories in the caption, so it makes sense to readers without having to refer to the methods. The blue category especially could benefit from a definition. For example, which categories are subsets of other categories (e.g., do yellow and green categories add up to the total in red?).

Our reply: We have added words describing each of color bars in the caption.

Fig 3 – I would encourage the authors/edits to print this image as large as possible in the published layout, so readers can make out the different countries.

Our reply: we have remapped this fig as large as possible in the published layout.

Fig 4 – Please convert axis number/name labels to English.

Our reply: Sorry for the mis-transformed. It has been modified and corrected.

Fig 5 – The regional labels are so spaced out that they are challenging to read. Please label each color individually rather than as a region (e.g., what region are the pale yellow and green?). I am

shocked to see the proportional trade sourced from Asia.

Our reply: Sorry for the mistake. We have remapped the fig to be clear.

Reviewer #2 (copy the lines and comments by authors)

Line 39, Would be good to establish right up front the difference b/w legal and illegal trade.

Our reply: We have added words to say that this including legal and illegal trade and cited two relevant references.

Line 59, Can you help the reader understand the justification for the need to resolve this issue?

Our reply: We have added a sentence to state the justification for the need to resolve this issue.

Line 77, Why does this matter? Why is this gap in knowledge problematic?

Our reply: We have provided justifications here.

Line 79, I don't find this document to be sufficiently helpful in understanding the methods for compiling the database. I am unclear as to the validity and reliability of the database. I am also not clear how the methods could be replicated—this is not to say they couldn't be, but the justification for the approach is unclear and how it could be replicated is unclear. For example, how were the lists comprised? Upon what justification are those lists sufficient?

All of this could be super useful b/c one day someone is going want to use this method for ornamental fish or other group of species.

Our reply: Very good comments. We appreciate very much that you think the methods useful for other taxa. We have taken these comments carefully, and described the methods in more detail to allow replication, and to give justifications for each step of methods. We have made changes as following:

1. For databases, we have added sentences to describe how we used CITES Trade database, LEMIS and ISIS and cited two additional publications that we followed to filter and clear the databases;
2. For online wildlife trade survey, we have described the online trade survey in more details. We have added two sentences to describe how we performed website search and cited an additional

reference; We have also added a paragraph to describe how we validated the crawled data and those in a website, and how we made-up missing listings in scraping. We have increased more words to describe how we obtained data on keys in more details.

3. For publications on historical online trade and physical markets. We have added sentences to describe how we used search phases to research the publications, and how we browse the publications. We have increased five sentences to describe the methods on collecting data on geographical ranges in more details.

We have a promise in the Reporting Summary submitted that data on the list of alien species is available on request, and all data will be published in a data paper in future. We have also added two new Supplementary Data (2 and 4) to show data on number of alien species and established species across countries or regions, which match with Fig 3 and 5.

Line 81, How do you know?

Our reply: We have deleted '(including illegal trade)'.

Line 84, So this makes sense to me although it is not justified with the literature.

So, then why present chord diagrams without countries as the unit of analysis?

Our reply: We have provided a reference for the definition here. We did not present chord diagrams based countries or regions (about 250 countries or regions) (Fig 6 and 7) because there are too many countries or regions. The figures based on countries or regions would be too complicated to understand (too many countries or regions and number of flows would be many times number of countries or regions. One even could not distinguish them clearly). We have made these maps following Capinha et al 2016 and Liu et al 2020 on invasive species.

Line 117, Do you feel all of these records are equally reliable? I can't image the data from Madagascar is on par with Australia...

Line 121, How do you know this is not just a function of detection probability? It makes sense to me b/c these are the countries with the best data collection, no?

Our reply: Very good questions. We have added one paragraph to clarify the sampling bias following this paragraph, and describe which countries have best data.

Line 125, The manuscript could be written in a much higher impact way, using the figures as illustrations and not wrote presentation of results.

Our reply: Very good suggestion! We have re-mapped the fig as you suggested, which show the values of alien species richness vs native species richness across 193 countries.

Line 152, Why? Upon what basis do you justify this approach?

Our reply: We have given the justifications for multimodel inference here.

Line 191, Isn't this somewhat tautological?

Our reply: We have rewritten the sentence.

Line 244, This is old now and there are newer data releases. If you can get access to newer data that would be awesome.

Our reply: Very good suggestion and we appreciate it very much. Following this comment, we have recollected the data on the most updated LEMIS data (1999-2020), and re-performed all analyses based on the new data. We also corrected errors in LEMIS data filtering in the previous version of the manuscript, which result in an overestimate of some alien richness values (mainly the USA and global).

Line 257, How do you justify the appropriateness of this QC?

Our reply: We re-edited this sentence.

Line 259, Such as?

Inclusion criteria for pets?

Date range?

How do you have confidence in these crawlers?

Our reply: Please see our reply to line 79.

Line 263, From which country? Google has different search functions in different countries?

Our reply: We have clarified here that we used Google Hong Kong.

Line 319, Can you describe these a bit more? Like how many?

How did you translate text?

Our reply: We have clarified these in more details.

Line 319, Do you have confidence in this list? Why or why not?

How do you justify putting legal and illegal trade together?

Our reply: Please see our reply to line 79. We have also added a sentence for justifications putting legal and illegal trade together.

Line 378, This seems like a really significant assumption, can you justify it?

Our reply: we have added some words to be specific for this, and please also see our reply to line 84.

Line 400, This is over a decade old now....seems like a lot has changed????

Our reply: Good point. We have recollected data on socio-economic factors in the recent 2015. Based on the new data collected, we have re-performed all analyses, and the results are similar.

Reviewer #3 (Remarks to the Author):

Overall, this paper is bringing together broader data on different types of animal invasives and WT, where before we mainly had smaller regional or taxonomic studies. In doing that it is addressing a research gap. However, the methods are a bit vague in places, and there are a lot of different data sources used, which are all hard to use even on their own, so I would need some more detail of the methods for things like the online sampling, literature search, and CITES analysis. The results are interesting but the discussion is also vague in some key places, and would benefit from more grounding in wider wildlife trade literature, better linking of conclusions to the findings, and more tangible recommendations for applications of these findings (e.g. rather than saying 'increase surveillance', give some actual recommendations for how to address this).

Our Reply: Thank you for considering our work as a research gap. We have taken your comments carefully and make changes as followings:

1. We have described the methods in more details. (I) for databases, we have added sentences to describe how we used CITES Trade database, LEMIS and ISIS and cited two additional publications that we followed to filter and clear the databases; (II) for online wildlife trade survey, we have described the online trade survey in more details. We have added two sentences to describe how we performed website search and cited an additional reference; We have also added a paragraph to describe how we validated the crawled data and those in a website, and how we made-up missing listings in scraping. We have increased more words to describe how we obtained data on keys in more details. (III) for publications on historical online trade and physical markets. We have added sentences to describe how we used search phases to research the publications, and how we browse the publications. We have increased five sentences to describe the methods on collecting data on geographical ranges in more details.
2. We have added discussions about results. We have added a paragraph to discuss the geographical patterns of alien species; We have also added several sentences to discuss the effects of area and population density on establishment richness. Moreover, we have added a paragraph to discuss differences in number of countries involved in trade and areas between established and unestablished species, and effects of species traits, market factors and other factors on establishment of trade species.

3. We have re-written the whole conclusions by linking our findings to applications, and giving more tangible recommendations for the applications from identifying current and future invasion hotspots to drafting regulations on release or escaping of exotic pets and early detection and rapid responses. We have also additionally cited several references for the reviewing wildlife trade.

Specific comments

L98 - I don't know what ISIS is

Our reply: We have added whole phases for CITES, LEMIS and ISIS here.

L101 - after all of this it is hard to know what 'these groups' means - I worked it out eventually but clarification would be helpful

Our reply: We have re-written the sentence to be clear.

L121 - as written this sounds like Western Europe is a country

Our reply: We have re-edited here.

L174 how does this reflect increasing complexities - this exchange pattern has been the case for some time and inter-continental trade is not an emerging issue

Our reply: You are correct. We have modified the sentence.

L193 I don't think you have provided enough to support this argument about range size - it reads like an add on that doesn't really link to your results

Our reply: Good comments. We have deleted these sentences related to discussion about range size.

L221 - what does 'surveillance' mean in this context? It would be useful to have a concrete recommendation here if you are saying it is urgent, as this is not clear.

L224 - not clear what you mean here about the CITES Trade Database - it is not designed to monitor alien species, especially as many potential invasives aren't listed on CITES, which you note next, but then why would it ever be expected to be sufficient for this? Surely Customs data would be more applicable?

L228 - the end of the paper doesn't really give clear ideas of what to do - it sounds a bit vague and I think this is a shame because the data have the potential to underpin some strong policy recommendations.

Our reply: Please see our Reply (3) to the general comments.

L236 the CITES TD is managed by UNEP not UNDP

Our reply: Sorry for the mistake. We have corrected this error.

L239 the CITES TD is a legal trade database - some Parties might report Source I as a seizure event but it is not that accurate to say it covers seizure data - see here for ways to ensure trade is not mischaracterized using these data <https://conbio.onlinelibrary.wiley.com/doi/full/10.1111/conl.12832>

Our reply: Good point. We have re-edited the sentence as you suggested and cited the reference.

L248 what purpose codes and other search terms did you use (your stated search of 'Live' on the CITES data would return every taxa listed as well as animals, including plants and taxa not in commercial trade).

Our reply: We have revised this part to clarify how we filter the data.

L262 The methods for the online trade sampling are not clear enough - does 'browse' mean somebody physically visited 95,000 websites?

Our reply: We have added two sentences to clarify how we browse the websites.

L345 what does 'intensively' searched mean? How did you select keywords? Did you follow any systematic online search protocols e.g. <https://www.cambridge.org/core/journals/oryx/article/systematic-survey-of-online-trade-trade-in-saiga-antelope-horn-on-russianlanguage-websites/3422BF90210A106E383E199D0E5D56CE> - if not please note how you ensured that the sampling was picking up as much as possible.

Our reply: Please see our reply (1) to the general comments.

Figure 1 - this looks really nice but is very hard to interpret (these multiple Venn diagrams always are!)

Our reply: We have remapped this fig and increase words to interpret the fig.

Figure 2 - the caption needs more detail

The maps - The silhouettes are a bit confusing where they are

Our reply: We have increased words to interpret in the caption of Fig 2-7. As some countries and regions in Fig 3 and 5 are too small to be seen clearly, we have additionally provided Supplementary Data 2 and 4 (excel files) to show the number of alien species and established species across each country or region for reader.

REVIEWER COMMENTS

Reviewer #3 (Remarks to the Author):

Checking this version was very difficult - there was no detailed response referring to line numbers, and the tracked changes version had so many changes (and no line numbers!) that it was close to impossible to find the relevant edits. Rather than writing 'we have changed the text' with no further information, it would be more transparent, and fair for the reviewers if this was done more clearly. I would strongly advise the authors to do a more detailed response next time, citing the new text in the response alongside a new line number.

I have been asked to check responses to both R2 and R3 and I do so below. For some of R2's comments, I could not work out what the original comment was about, or whether it had been addressed due to the issues noted above.

R2

- the authors have not addressed the comment about the difference between legal and illegal trade (see original comment for L59).
- I don't think COVID-19 is the only justification for this (see original comment for L59) - I would expand upon it.
- Original comment "Line 191, Isn't this somewhat tautological? Our reply: We have rewritten the sentence." - I cannot tell whether this has been addressed.
- Original comment "Line 319, Can you describe these a bit more? Like how many? How did you translate text? Our reply: We have clarified these in more details. - I cannot tell whether this has been addressed.
- Original comment Line 378, 'This seems like a really significant assumption, can you justify it? Our reply: we have added some words to be specific for this, and please also see our reply to line 84'. - I cannot tell whether this has been addressed.

R3

- The authors have addressed my comments on definition of acronyms, and ambiguous language well.
- The methods and database search terms are now better described but please report all search terms used for the CITES database search so that somebody could recreate this in the future - you say you only specified taxonomic groups and 'live', but did you use all term, source, unit, purpose codes - e.g. did it matter whether it was 'trade' or not - as this will include species not in commercial trade. Your paper seems very focussed on 'trade', but this will include captive bred animals being exchanged between zoos too, or movement of personal pets, and I can't tell whether that counts for your aims.
- Also did you just look for live trade, or anything that could be viable, like live eggs?
- I can't refer back to your tracked changes version because there are no line numbers, but the bit that says "Not all animals in zoos are sourced from trade, and threatened species (categorized as vulnerable, endangered, or critically endangered) in zoos are being bred for ex-situ conservation or conservation campaign 58. We therefore collected data on scientific name, class and family of animals kept in zoos and countries from ISIS (ISIS.IUCN.Matching.xls, containing 94,877 records of mammals, birds, reptiles, and amphibians) but excluding threatened species." I do not understand why threatened species have been excluded - this justification seems to be based on a big assumption. Did you exclude threatened species from any of the other databases? According to your methods you haven't excluded zoo exchanges from the CITES data (see above), so why would you exclude potential captive breeding species from the zoo data?

Point-by-Point Responses to New Comments From Reviewer #3

Reviewer #3:

Checking this version was very difficult - there was no detailed response referring to line numbers, and the tracked changes version had so many changes (and no line numbers!) that it was close to impossible to find the relevant edits. Rather than writing 'we have changed the text' with no further information, it would be more transparent, and fair for the reviewers if this was done more clearly. I would strongly advise the authors to do a more detailed response next time, citing the new text in the response alongside a new line number.

I have been asked to check responses to both R2 and R3 and I do so below. For some of R2's comments, I could not work out what the original comment was about, or whether it had been addressed due to the issues noted above.

Our reply: We are very sorry for not providing line numbers in the revised version, and responses referring to line numbers in the letter. In this revision, we have rectified this, including line numbers in updated versions, and this letter, and marking where we have revised.

R2

- the authors have not addressed the comment about the difference between legal and illegal trade (see original comment for L59).

Our reply: We have added a sentence to address the difference between legal and illegal trade (line 184-188, line 193 in current tracking version).

- I don't think COVID-19 is the only justification for this (see original comment for L59) - I would expand upon it.

Our reply: You are right. We have added more justifications to expand on this (line 222-224 in current tracking version)

- Original comment "Line 191, Isn't this somewhat tautological? Our reply: We have rewritten the sentence." - I cannot tell whether this has been addressed.

Our reply: We have revised this line (line 1105-1106). The original sentence is “These findings suggest that aliens dominate species richness in the live wildlife trade, reflecting increased globalization of exotic pets in trade ⁷⁻⁹.” We have revised this sentence by deleting “ , reflecting increased globalization of exotic pets in trade ⁷⁻⁹”.

- *Original comment “Line 319, Can you describe these a bit more? Like how many? How did you translate text? Our reply: We have clarified these in more details. - I cannot tell whether this has been addressed.*

Our reply: We have described these in more detail and provided number of websites with PDF file (line 1730-1748). Original sentence is “We crawled all websites of wildlife trade except websites displaying listings in PDF. In this case, we directly downloaded PDF files and transferred them into the text”. We have revised it as “We crawled all websites relating to wildlife trade, except for one website that displayed its listings in PDF format. In this case, we directly downloaded the PDF file and copied information from the PDF file into the text.”

- *Original comment Line 378, ‘This seems like a really significant assumption, can you justify it? Our reply: we have added some words to be specific for this, and please also see our reply to line 84’. - I cannot tell whether this has been addressed.*

Our reply: Yes, this was not addressed clearly in the previous version. We are sorry for this. In the current version, we have added more detailed text to clarify how we matched the combinations of a traded species and traded countries or regions with those of the species and native countries or regions using vlookup function (line 1879-1910), including detailed definitions of each variable in the function here (line 1882-1885), and specifying the meanings of matched and unmatched combinations (line 1885-1910). We have also re-edited some sentences to address that we identified alien species in global administrative areas at country or region level (line 390-391, 1855-1856), which provides justification for why we matched the combinations of species and countries or regions.

R3

- *The authors have addressed my comments on definition of acronyms, and ambiguous language well.*

Our reply: Thank you for positive comments on revised version of the manuscript.

- The methods and database search terms are now better described but please report all search terms used for the CITES database search so that somebody could recreate this in the future - you say you only specified taxonomic groups and 'live', but did you use all term, source, unit, purpose codes - e.g. did it matter whether it was 'trade' or not - as this will include species not in commercial trade. Your paper seems very focussed on 'trade', but this will include captive bred animals being exchanged between zoos too, or movement of personal pets, and I can't tell whether that counts for your aims.

Our reply: We have added text to describe in more detail how we collected data from CITES and LEMIS (line 1472-1522). As both CITES and LEMIS are large trade databases, we treated each transaction in the two databases as a trade record, following work by Marshall and colleagues (2021), and added this citation. We also have added descriptions of trade purposes, and units for live trade, and how we considered trade volume.

- Also did you just look for live trade, or anything that could be viable, like live eggs?

Our reply: We have added a sentence to clarify that we just look for live trade, and did not consider "eggs" as live in this case and gave the justification (line 1518-1520).

- I can't refer back to your tracked changes version because there are no line numbers, but the bit that says "Not all animals in zoos are sourced from trade, and threatened species (categorized as vulnerable, endangered, or critically endangered) in zoos are being bred for ex-situ conservation or conservation campaign 58. We therefore collected data on scientific name, class and family of animals kept in zoos and countries from ISIS (ISIS.IUCN.Matching.xls, containing 94,877 records of mammals, birds, reptiles, and amphibians) but excluding threatened species." I do not understand why threatened species have been excluded - this justification seems to be based on a big assumption. Did you exclude threatened species from any of the other databases? According to your methods you haven't excluded zoo exchanges from the CITES data (see above), so why would you exclude potential captive breeding species from the zoo data?

Our reply: Good questions, which would improve the quality of the manuscript. We have considered the questions very carefully, and made the following revisions. We have started a new paragraph to

clarify how we collected data from ISIS in more detail (line 1523-1548). We have added a sentence to describe the ISIS data in more detail (line 1523-1532) - these have no transaction records among zoos or institutions, making us difficult to identify the species involved in trade. We have clarified that the inclusion of all threatened species from ISIS into our database would bring problems of overestimation of traded species (line 1532-1533). We have also provided justifications for why we excluded threatened species in ISIS, citing the work Conde and colleagues (2014) (line 1533-1539). We have provided an additional table (Table S18 in supplementary materials) showing how many threatened species from ISIS were excluded, and give the number of these species that are not excluded in other databases, to clarify the actual picture of the data excluded in the whole dataset (line 1539-1543). We have added sentences to show the weak effects of such exclusions on total numbers of species in our databases (line 1543-1548).

REVIEWERS' COMMENTS

Reviewer #1 (Remarks to the Author):

The authors have responded to my and Reviewer #2's questions and suggestions comprehensively, including with rigorous additional data analysis and discussion. The paper – including methods and discussion with policy/conservation applications – is significantly more clearly written. The study overall reports an important analysis and would make a meaningful contribution to the field of wildlife trade and biodiversity conservation.

Below are a few final minor suggestions:

L41 The point of the first phrase of the first sentence is unclear. I think the authors' point in that sentence is that wildlife trade represents a high monetary value (hence representing a meaningful human commodity and a significant contribution to the global economy?) but also represents significant threats. The first part of the phrase would be clearer if the authors could just clarify what they are insinuating with the high monetary value (important human commodity or contribution to global economic or something else?).

For simplify, conciseness and ease of reading, I recommend combining figures that have similar symbology and thus captions: Fig 3 and 5 together, and Fig 6 and 7 together, each with parts (a) and (b).

Reviewer #3 (Remarks to the Author):

The comments have been addressed well and I think the paper is a nice contribution to the literature.

A point-by-point response to the reviewers' comments

Reviewer #1 (Remarks to the Author):

The authors have responded to my and Reviewer #2's questions and suggestions comprehensively, including with rigorous additional data analysis and discussion. The paper – including methods and discussion with policy/conservation applications – is significantly more clearly written. The study overall reports an important analysis and would make a meaningful contribution to the field of wildlife trade and biodiversity conservation.

Our reply: Thank you for your overall positive comments on our revisions

Below are a few final minor suggestions:

L41 The point of the first phrase of the first sentence is unclear. I think the authors' point in that sentence is that wildlife trade represents a high monetary value (hence representing a meaningful human commodity and a significant contribution to the global economy?) but also represents significant threats. The first part of the phrase would be clearer if the authors could just clarify what they are insinuating with the high monetary value (important human commodity or contribution to global economic or something else?).

Our reply: Thank you for your suggestions. We have revised line 41 to integrate the suggestions by the reviewer into the sentence (line 41-43)

For simplify, conciseness and ease of reading, I recommend combining figures that have similar symbology and thus captions: Fig 3 and 5 together, and Fig 6 and 7 together, each with parts (a) and (b).

Our reply: We have combined Fig 3 and Fig 5 into Fig 3 with parts (a) and (b), and Fig 6 and 7 into Fig 5 with parts (a) and (b). Yes, they look more conciseness and ease for reader.

Reviewer #3 (Remarks to the Author):

The comments have been addressed well and I think the paper is a nice contribution to the literature.

Our reply: Thank you for your satisfying our revisions and positive comments on our manuscript.